# Metformin Inefficiency to Lower Lipids in Vitamin B12 Deficient HepG2 Cells Is Alleviated via Adiponectin-AMPK Axis

**DOI:** 10.3390/nu15245046

**Published:** 2023-12-08

**Authors:** Joseph Boachie, Victor Zammit, Ponnusamy Saravanan, Antonysunil Adaikalakoteswari

**Affiliations:** 1Division of Metabolic and Vascular Health, Clinical Sciences Research Laboratories, Warwick Medical School, University of Warwick, University Hospital-Walsgrave Campus, Coventry CV2 2DX, UK; jboachie@ucc.edu.gh (J.B.); v.a.zammit@warwick.ac.uk (V.Z.); p.saravanan@warwick.ac.uk (P.S.); 2Diabetes Centre, George Eliot Hospital NHS Trust, College Street, Nuneaton CV10 7DJ, UK; 3Populations, Evidence and Technologies, Division of Health Sciences, Warwick Medical School, University of Warwick, Coventry CV4 7HL, UK; 4Department of Bioscience, School of Science and Technology, Nottingham Trent University, Nottingham NG11 8NS, UK

**Keywords:** lipid metabolism, non-alcoholic fatty liver disease (NAFLD), fatty acid oxidation (FAO), lipogenesis, mitochondria, obesity

## Abstract

**Background:** Prolonged metformin treatment decreases vitamin B12 (B12) levels, whereas low B12 is associated with dyslipidaemia. Some studies have reported that metformin has no effect on intrahepatic triglyceride (TG) levels. Although AMP-activated protein kinase (AMPK) activation via adiponectin lowers hepatic TG content, its role in B12 deficiency and metformin has not been explored. We investigated whether low B12 impairs the beneficial effect of metformin on hepatic lipid metabolism via the AMPK-adiponectin axis. **Methods**: HepG2 was cultured using custom-made B12-deficient Eagle’s Minimal Essential Medium (EMEM) in different B12-medium concentrations, followed by a 24-h metformin/adiponectin treatment. Gene and protein expressions and total intracellular TG were measured, and radiochemical analysis of TG synthesis and seahorse mitochondria stress assay were undertaken. **Results**: With low B12, total intracellular TG and synthesized radiolabelled TG were increased. Regulators of lipogenesis, cholesterol and genes regulating fatty acids (FAs; TG; and cholesterol biosynthesis were increased. FA oxidation (FAO) and mitochondrial function were decreased, with decreased pAMPKα and pACC levels. Following metformin treatment in hepatocytes with low B12, the gene and protein expression of the above targets were not alleviated. However, in the presence of adiponectin, intrahepatic lipid levels with low B12 decreased via upregulated pAMPKα and pACC levels. Again, combined adiponectin and metformin treatment ameliorated the low B12 effect and resulted in increased pAMPKα and pACC, with a subsequent reduction in lipogenesis, increased FAO and mitochondrion function. **Conclusions**: Adiponectin co-administration with metformin induced a higher intrahepatic lipid-lowering effect. Overall, we emphasize the potential therapeutic implications for hepatic AMPK activation via adiponectin for a clinical condition associated with B12 deficiency and metformin treatment.

## 1. Introduction

The global epidemic of obesity is increasing, and evidence from more than seventy countries shows that the prevalence has doubled in nearly four decades [1]. Likewise, the incidence of obesity-related complications such as non-alcoholic fatty liver disease (NAFLD) is increasing proportionately. The worldwide prevalence of NAFLD in 2022 was shown to be 32.4% [2], and about 3% of all deaths in Europe in recent years were reported to be liver-related [3]. A higher risk of gestational diabetes has been evidenced in pregnant women with vitamin B12 (B12) deficiency [4]. Maternal low B12 was shown to be independently associated with adverse cord blood lipids [5] and with elevated serum cholesterol and triglycerides in adults [6]. In a C57BL/6 mice study, B12 deficiency accounted for an adverse adipose lipid profile [7]. Our studies in adipocytes demonstrated B12-induced reduction in methylation potential and hypo-methylation at promoter regions of low-density lipoprotein receptor (LDLR) and sterol regulatory element binding protein (SREBF1) genes [8]. Likewise in hepatocytes, we provided evidence of low B12-induced dysregulation of intrahepatic fatty acid metabolism [9]. The risk of low B12 levels increases with higher metformin dose and longer treatment duration, as well as in patients with risk factors for vitamin B12 deficiency [10].

Metformin, the current first-line therapy globally for management of T2D, has been thought to be beneficial in reducing fat accumulation in the liver. It has been widely accepted that metformin activates AMP-activated protein kinase (AMPK) via inhibition of lipogenesis [11], elevation of hepatic FAO [12] and improvement of mitochondrial function [13]. However, few animal studies and many human studies have shown that metformin treatment had no effect on intra-hepatic TG content [14]. Also, there are restrictions in metformin-induced activation of AMPK, suggesting that AMPK might be regulated by other metabolites [15]. Metformin is also known to reduce adiponectin in adipocytes involving activation of AMPK. Adiponectin, an adipocytokine, negatively correlates with adiposity and body mass index (BMI) [16]. Adiponectin exerts it biological effect via adiponectin receptors (AdipoR1/R2) via AMPK to potentially regulate the hepatic metabolism of lipids and inhibit lipogenesis genes (FAS and ACC), leading to reversal of NAFLD [17]. Similarly, adiponectin has been shown to upregulate FAO by increasing carnitine palmitoyl transferase 1 (CPT1) activity [18] as well as mitochondrial number and function [19]. However, the receptors of adiponectin representing probable targets of therapy in obesity-oriented disorders have been demonstrated to be decreased in conditions of obesity and diabetes [20], and the mechanisms are largely unknown.

Prospective studies have shown that not only do metformin users have lower B12 levels, they also show no change in FA metabolism [21]. Animal data suggest metformin does not prevent NAFLD from developing into more severe NASH or fibrotic stages [22]. Prolonged metformin treatment depleting hepatic stores of B12 could result in lipid dysregulation. This metabolic overburden in the liver might be responsible for the ineffectiveness of the long-term metformin treatment; however, the mechanisms are unknown. Therefore, to understand the regulation underlying the clinical condition associated with long-term metformin treatment and to enhance the therapeutic potential, we aim to investigate whether (1) low B12 impairs the action of metformin on hepatic lipid metabolism and (2) activation of the AMPK-adiponectin axis mitigates the metformin action with low B12 status.

## 2. Experimental Procedures

**Cell culture:** Briefly, culture of the HepG2 cell line was carried out in T75-flasks and seeded in 6-well plates with custom-made EMEM medium supplemented with different B12 concentrations, namely, 500 nM (control), 1 nM (1000 pM), 100 pM and 25 pM (low B12), as previously described [9] (details of HepG2 source, authenticity and mycoplasma assays have been provided in Appendix A). After the last medium change in plates with cells at 90–100% confluence, the EMEM-B12 medium was then replaced with a serum-free medium (0.1% BSA, 1% L-Glutamine) overnight. The cells were maintained for 24 h in either a serum-free medium only, serum-free medium supplemented with either metformin (1 mM and 2 mM) only, adiponectin (25 ng) only or a combination of adiponectin (25 ng) and metformin (2 mM) (1 mM AICAR as positive control). The cells were subsequently harvested and stored at −20 °C until required for RNA and protein assays. The same treatments were repeated in HepG2 cells and utilized for radio-chemical analysis in 6-well plates and extracellular seahorse XF24 flux assay using 24-well seahorse plates, respectively.

**RNA isolation, cDNA synthesis and gene expression:** Total RNA isolation was carried out using the Trizol method. cDNA synthesis and gene expression assays were same as previously described [23]. The qRT-PCR involved the use of 18s rRNA (Applied Biosystems, Knutsford, UK) for normalizing expressions of RNA. All fatty acid, triglyceride, cholesterol biosynthesis and fatty acid oxidation (FAO) genes were custom taqman gene expressions supplied by Applied Biosystems, Knutsford, UK.

**Western blot analysis:** HepG2 cells in plates were washed twice with ice-cold phosphate-buffered saline (PBS) followed by harvest using a lysis buffer radioimmunoprecipitation assay (RIPA) with dissolved phosphatase and protease inhibitors and placed in storage at −80 °C until usage. Protein samples were subsequently quantified using a Bradford assay. Running and analysis of the Western blots were performed as described previously [24].

**Total intracellular TG estimation:** Estimation of total intracellular TG in the cells was performed using the commercial triglyceride quantification kit (ab65336) from Abcam, Cambridge, UK as previously described [9]. Briefly, reconstitution and preparation of the TG standard, probe, enzyme mix and lipase were carried out following the manufacturer’s protocol. The TG standard was subsequently diluted with the assay buffer to a total volume of 50 μL per well to generate standard concentrations of 0, 2, 4, 6, 8 and 10 nmol/well, respectively. Also, 2 μL of lipase was added to each well and incubated for 20 min. A cocktail of reaction mix (50 μL) comprising TG assay buffer, probe and enzyme mix (46:2:2) was added to each well and gently mixed and the absorbance was measured at a wavelength of 570 nm. The data was used to derive a standard curve. Afterwards, the samples were prepared by homogenising cells in PBS and subsequently making up cell suspensions to 50 μL of TG assay buffer per well. A 2 μL quantity of lipase was then added and incubated for 20 min at room temperature, followed by addition of 50 μL of the reaction mix. The final content was gently mixed, and the absorbance measured at 570 nm. The concentration of TGs was calculated from the standard curve.

**Radiochemical assay of synthesized TG:** HepG2 cells were seeded in 6-well plates in different concentrations of B12 until confluence was reached, followed by incubation with ^14^C-Oleate (0.75 mM concentration at 2000 dpm/nmol) and L-carnitine (1 mM) to facilitate incorporation of ^14^C-Oleate into hepatocytes for 2 h in EMEM, followed by a 5-min incubation for background normalizing control at 37 °C and 5% CO_2_ saturation. The cells were harvested in 2 mL methanol, and the synthesized radiolabelled-TG was extracted using a mobile phase, such as hexane/diethyl ether/formic acid (*v*/*v*/*v*, 70/30/1), on a thin-layer chromatography (TLC) plate with 10 nmol glyceryltripalmitate (tripalmitin) as a standard. The TG bands on the TLC plates were transferred into vials containing 5 mL scintillation fluid for scintillation counter-assessment of radioactivity and normalized per milligram (mg) protein, as described previously [9].

**Seahorse extracellular flux assay of mitochondrial dysfunction:** The culture of HepG2 cells in T-75 flasks in different B12 concentrations of EMEM media, seeding of HepG2 in XF24 seahorse plates and preparation of different concentrations of seahorse inhibitors for the seahorse assay were same as in our previous study [9]. **Sample preparation and seahorse assay:** After 24 h incubation at 37 °C and 5% CO_2_ saturation, the cells in XF24 seahorse plates were washed in KHB buffer by removing 200µL EMEM medium off the cells, followed by addition of 1 ml of KHB. A 950 µL quantity of the KHB in 24-well seahorse plates was then removed, followed by addition of another 675 µL of KHB to the wells of the plate. The plates were then incubated at 37 °C without CO_2_ saturation for 1 h. To run the seahorse assay, the cartridge was first loaded into the seahorse analyser for calibration and equilibration according to the manufacturers’ protocol. The utility plate was then replaced with the seahorse XF24 plate, and oxygen consumption rates (OCRs) in cells were measured as similarly described [25]. **Maximal respiratory capacity/oxygen consumption rate (OCR):** The maximal respiratory capacity in HepG2 involving the assessment of oxygen consumption rate (OCR) of cells in a rich substrate medium (glucose 2.5 mM, pyruvate 1 mM, L glutamin 2 mM and BSA 0.1%) was analysed. Likewise, the respiratory capacity in a limited substrate or high palmitate supply medium (0.5 mM L-carnitine and 1.25 mM glucose) was measured using the Seahorse 24 XF flux analyser, as described previously [9].

**Statistical analysis:** Analysis of all data was performed using the software program Prism 8 (GraphPad, San Diego, CA, USA). All quantitative measurements, where applicable, were obtained in triplicate for standards, controls and cases to ensure precision of data. Data obtained from total intracellular triglycerides assay, radio-chemical assay and seahorse flux assays were normalized with the total protein concentration (mg) in each sample (n) of Hep G2 cells to alleviate possible variations that might misrepresent the data. The final data were then expressed as mean + standard error of mean (SEM). Also, data for all samples obtained were tested for normality prior to analysis using D’Agostino–Pearson omnibus and Shapiro–Wilk normality tests. For samples that passed the normality assessment, comparisons between cases and their corresponding controls were carried out using a two-tailed unpaired *t*-test. Likewise, where the test for normality was not passed, comparisons between two sample groups were carried out using the 2-tailed Mann–Whitney U test. Statistically significant differences were assigned *p*-values < 0.05.

## 3. Results

### 3.1. Low B12 Impedes Metformin Action on Intracellular TG and Expression of Genes in Fatty Acid Synthesis

To assess whether low B12 interferes with metformin treatment, we quantified the total intracellular TG in metformin-treated cells under different B12 conditions. In the absence of metformin (0-mM), the total intracellular TG level was significantly higher with low B12 compared to controls. Also, metformin treatment in control cells showed significant reduction (20.7%) of TG levels, but treatment in low-B12 cells did not achieve a significant reduction (6.2%) (Figure 1A). Then we performed a radiochemical analysis by exposing the cells to radio-labelled fatty acid (^14^C-Oleate) and assessed the amount of radio-labelled TG synthesized. In the absence of metformin (0-Mm), we observed that the level of TG synthesized was significantly higher in low-B12 cells compared with controls. However, in metformin-treated cells, the level of TG synthesized was significantly reduced in the control (58%) but not in low-B12 cells (13%) (Figure 1B). These data confirm that metformin does not have a significant reduction effect on the intracellular TG level and esterification of FAs for TG synthesis with low B12.

To assess whether metformin effectively lowers FA synthesis under different B12 conditions, we analysed the mRNA expression of various genes regulating the FA synthesis pathway. In the absence of metformin (0-mM), there was a significantly higher expression of the genes regulating FA synthesis with low B12 compared to controls. In the control cells with metformin treatment, there was greater reduction in the expression of genes, including the nuclear transcription factor and master regulator of lipogenesis—sterol regulatory element binding protein (SREBF1) (49.6%), ATP citrate lyase (ACLY) (21.0%), acetyl CoA carboxylase (ACC) (23.8%), fatty acid synthase (FASN) (36.0%) and elongation of very-long fatty acid (ELOVL6) (35.8%) (Figure 1C(i–v)). Also, in low-B12 cells with metformin treatment, SREBF1 was likewise significantly decreased (34%). We also observed a significantly decreased expression of FA synthesis genes such as ACLY (17.6%), FASN (27.4%) and ELOVL6 (29.9%). However, the expression of the rate-limiting enzyme, ACC, was not significantly reduced in metformin-treated cells with low B12 (8.9%), compared with controls (Figure 1C(i–v)). Although metformin showed significant reduction in FA genes (except ACC) with low B12, these data suggest that the lowering effect of metformin on the FA synthesis in control was greater than in the low B12 hepatocyte cell line. Furthermore, we observed that in the absence of metformin (0-mM), the expression of TG biosynthesis genes including SCD1, GPAM, AGPAT, DGAT1 and DGAT2 was significantly higher in low-B12 cells. However, following metformin treatment in control cells, the expression of the TG biosynthesis genes SCD1 (36.3%), GPAM (26.1%), AGPAT (40.5%), DGAT1 (48.2%) and DGAT2 (29.0%), and in low-B12 cells SCD1 (22.3%), GPAM (21.1%), AGPAT (39.5%), DGAT1 (37.1%) and DGAT2 (23.8%) was significantly lower (Figure 1D(i–v)). Here the findings again suggest that metformin decreases gene expression of TG synthesis significantly in both control and low-B12 cells, but more in control cells than in those with low B12. We next assessed the lowering effect of metformin on cholesterologenesis. In the absence of metformin (0-mM), the mRNA expressions of HMGCR, HMGCS1 and LDLR were significantly higher than in low-B12 cells. However, in control cells with metformin treatment, we observed a significantly decreased expression of HMGCR (32.9%), HMGCS1 (32.6%) and LDLR (34.2%), with similar observation with low B12, which had decreased expression of HMGCR (24.4%), HMGCS1 (27.1%) and LDLR (27.6%). (Figure 1E(i–iii)). The findings suggest that metformin significantly decreases expression of genes in cholesterol synthesis in both control and low-B12 cells, but greater in control cells than low B12.

### 3.2. Low B12 Impedes Metformin Action on Fatty Acid Oxidation (FAO) and Mitochondria Function via Impaired AMPK

We assessed whether metformin was effective in upregulating FAO in hepatocytes with low B12. In the absence of metformin (0-mM), we observed that the expressions of CPT1a, CACT, HADHA, ACADS, ACADM and ACADL were significantly decreased in cells with low B12 compared with controls. However, in control cells with metformin treatment, there was a significant increase in the expression of CPT1a (35.5%), CACT (37.8%), HADHA (52.0%), ACADS (79.3%), ACADM (74.3%) and ACADL (27.9%) (Figure 2B). Contrarily, there was no significant increase in CPT1α (2.5%), CACT (14.7%), HADHA (14.1%), ACADS (29.6%), ACADM (23.6%) and ACADL (2.0%) in metformin-treated hepatocytes with low B12 (Figure 2A(i–vi)). This evidence suggests that in low-B12 hepatocytes, the expression of FAO genes is lower and their levels following metformin treatment were not significantly increased, therefore accounting for impaired FAO.

FAO is functionally interconnected with mitochondrial respiration involving the electron transport chain. Since FAO was impaired in metformin-treated hepatocytes with low B12, we further assessed this effect on mitochondrial functional efficiency about the utilization of substrate supply for mitochondrial respiration with low B12 following metformin treatment. First, we assessed whether B12 independently affects mitochondrial functional integrity in the utilization of substrate supply for energy metabolism, using the seahorse XF24 flux mito-stress assay. In the absence of metformin, we observed that the spare respiratory capacity (SRC) of the mitochondria in hepatocytes with low B12 was significantly lower compared with controls (Figure 2B(i)). This suggested that low B12 independently impaired mitochondrial functional potency in hepatocytes by reducing spare respiratory capacity. Afterwards, we assessed whether metformin improves mitochondrial functional efficiency in low-B12 hepatocytes. We observed a significant increase (43.6%) in the SRC of metformin-treated hepatocytes in controls, whereas upregulation of SRC was not significant with low B12, accounting for only an 8.6% increase (Figure 2B(ii)), therefore suggesting that restoration of mitochondrial respiratory function via metformin treatment was impaired in hepatocytes with low B12 compared with controls. Subsequently, we assessed whether metformin efficiently improves mitochondrial respiration by utilizing long-chain FA palmitate with low B12. The upregulation of SRC in metformin-treated cells in palmitate-rich substrate was significantly impaired with low B12 (17.0%) compared with controls B12 (60.0%) (Figure 2B(iii)), therefore suggesting that mitochondrial functional efficiency in metformin-treated hepatocytes was compromised with low B12, resulting in the inefficient utilization of palmitate for respiration. This may hence account for the accumulation of long-chain FA (palmitate) in metformin-treated hepatocytes with low B12, leading to dyslipidaemia.

Metformin is known to achieve a lowering effect on lipids via the activation of AMPK and subsequent upregulation of FAO. Hence, we assessed the independent effect of B12 on pAMPKα and pACC levels in hepatocytes to ascertain whether B12 affects the activation of AMPK with subsequent inactivation of ACC. We observed that pAMPKα level in HepG2 cells was significantly decreased with low B12 compared with controls (Figure 2C(i)). Similarly, the level of pACC was significantly decreased with low B12 compared with controls, concurrent to gene expression (Figure 2C(ii)). Our observation suggests that the phosphorylation (activation) of AMPK and subsequent phosphorylation (inactivation) of ACC were decreased with low B12 compared with controls. Since the levels of pAMPKα and pACC were significantly reduced with low B12, we further assessed whether the activation of AMPK and subsequent inactivation of ACC would be restored in low-B12 hepatocytes following metformin treatment. We observed that the levels of pAMPKα and pACC (Figure 2D(i,ii)) increased significantly (93.1% and 115.6%, respectively) in control cells treated with metformin but not significantly (22.4% and 37.7%, respectively) in metformin-treated cells with low B12. This evidence suggests that metformin did not efficiently restore the levels of pAMPKα and pACC with low B12, therefore confirming that impaired FAO may be mediated via AMPK in low-B12 metformin-treated hepatocytes.

### 3.3. Adiponectin Ameliorates the Regulation of FA Synthesis in Low B12 Hepatocytes via Increased AMPK Activation and AdipoR1/R2

To assess the role of adiponectin in the hepatic metabolism of lipids, we tested for the expression of adiponectin and its receptors in hepatocytes. We observed that adiponectin was not expressed in HepG2; however, expression of adiponectin receptors (AdipoR1 and AdipoR2) was detected. Therefore, we assessed the expression of these receptors in the cells under different B12 conditions and observed that the mRNA expression of both AdipoR1 and R2 were decreased with low B12 compared to controls. (Figure 3A(i,ii)). Adiponectin is proposed to induce the activation of signalling pathways, leading to regulation of lipid metabolism by binding to adipoR1 and R2 receptors in the liver. We next assessed whether treatment with adiponectin would affect receptors in HepG2. Following adiponectin treatment, in control cells, we observed a significant increase in the expression of adipoR1 (43.5%) and R2 (45.0%), as well as similarly increased adipoR1 (67.7%) and R2 (81.0%) receptors, with low B12 (Figure 3B(i,ii)). This implies that the adiponectin treatment upregulates adipoR1 and R2 receptors significantly with low B12.

We further assessed whether adiponectin treatment and subsequent upregulation of adipoR1/R2 affect the activation of AMPK in B12 conditions. We observed that the levels of pAMPKα and pACC were significantly lower with low B12 compared with controls. However, pAMPKα and pACC levels as observed in control cells (199.6% and 83.3%-increase) were significantly increased with low B12 (426% and 211%, respectively) following adiponectin (positive control-AICAR) treatment (Figure 3C(i,ii)). Therefore, this suggests that there was significant activation (phosphorylation) of AMPKα and subsequent inactivation (phosphorylation) of ACC in low-B12 hepatocytes following adiponectin treatment.

Next, observing the significant elevation in the levels of pAMPKα and pACC in low-B12 cells following adiponectin treatment, we assessed this treatment on several genes regulating lipogenesis and FAO in hepatocytes. In basal condition (without adiponectin), we observed that the gene expression of the key nuclear transcription factor SREBF1 and downstream genes regulating FA synthesis (ACC, FASN and ELOVL6) was upregulated in low-B12 hepatocytes. However, treatment with adiponectin (positive control—AICAR) in control cells resulted in significantly decreased expression of SREBF1 (22.1%), ACC (41.1%), FASN (36.8%) and ELOVL6 (22.2%). Likewise, in low-B12 cells, significantly decreased expression of SREBF1 (41.5%), ACC (49.3%), FASN (46.7%) and ELOVL6 (44.6%) was observed (Figure 3D(i–iv)). This therefore suggests that adiponectin was effective in reducing FA synthesis in low-B12 hepatocytes. We observed that the gene expression of enzymes regulating TG synthesis (SCD1, DGAT1 and DGAT2) and the master regulator of cholesterol (LDLR) biosynthesis under basal conditions was increased with low B12 compared with controls. However, treatment with adiponectin (positive control AICAR) in resulted in significant reduction of expression of genes—SCD1 (27.2%), DGAT1 (26.3%), DGAT2 (23.1%) and LDLR (24.1%) in control cells. Similarly, in low-B12 hepatocytes, significantly decreased expression of genes—SCD1 (39.8%), DGAT1 (39.1%), DGAT2 (41.4%) (Figure 3E(i–iii)) and LDLR (36.4%) (Figure 3F)—was observed. Since adiponectin was effective in the reduction of gene expression of lipid (FA, TG and cholesterol) synthesis, we further assessed its effect on the total intracellular TG levels in low-B12 hepatocytes. Similarly, we observed that total intracellular TG level under basal conditions was initially higher with low B12 compared with controls. Following treatment with adiponectin (positive control—AICAR), the total TG level with low B12 was significantly reduced by 33.2% compared with hepatocytes in control B12 (20.5%) (Figure 3G). This suggests that adiponectin reduces intracellular TG levels significantly in hepatocytes with low B12. To confirm adiponectin effect on TG synthesis, we performed a radiochemical assay to assess the uptake and esterification of FA (radiolabelled) for TG synthesis in low-B12 hepatocytes. Under basal conditions, the exposure of HepG2 to radiolabelled-FA (^14^C-Oleate) resulted in significantly high levels of TG with low B12 compared with controls. However, adiponectin (positive control—AICAR) treatment resulted in significantly lower levels of TG synthesized with low B12 (41.9%) compared with controls (23.6%-decrease) (Figure 3H). This therefore shows that adiponectin treatment alleviates the low B12-induced increased uptake and esterification of FA for TG synthesis in low-B12 hepatocytes.

### 3.4. Adiponectin Ameliorates the Regulation of FAO in Low B12 Hepatocytes via Increased AMPK Activation

Since adiponectin decreased lipogenesis significantly in low-B12 cells, we further assessed how FAO pathway might be affected by adiponectin treatment. Under basal (without adiponectin) conditions, the gene expression of CPT1α and ACADM regulating FAO pathway was decreased in low-B12 hepatocytes compared with control B12. However, adiponectin (positive control—AICAR) treatment resulted in significant upregulation of FAO genes with low B12 accounting for 60.8% increase in CPT1α and 56.6% in ACADM compared with controls (CPT1—37.2% and ACADM—47.6%—increase) (Figure 4A(i,ii)).

Next, we assessed the independent effects of adiponectin on mitochondrial functional efficiency in the utilization of substrate supply for respiration. We observed that under basal (without adiponectin) conditions, functional efficiency (OCR and SRC) was decreased with low B12 compared with controls. However, treatment with adiponectin (positive control—AICAR) resulted in significant increase in mitochondrial functional efficiency (OCR and SRC) with low B12 (89% increase) compared with basal control (42%). This evidence indicates that there was significant improvement in functional efficiency induced by adiponectin in hepatocytes with low B12 (Figure 4B(i,ii)). Furthermore, we assessed the effect of adiponectin on the uptake of long-chain FA (palmitate) from the substrate medium for mitochondrial respiration in HepG2. The functional efficiency (OCR and SRC), following adiponectin (positive control—AICAR) treatment, was significantly increased in cells exposed only to palmitate with low B12 (122%) compared to controls (68%) (Figure 4C(i,ii)). Likewise, exposure to palmitate in the presence of respiratory inhibitors (oligomycin, FCCP and rotenone/antimycin) accounted for 108% increase with low B12 compared with controls (120%) (Figure 4D(i,ii)). This evidence signifies that adiponectin enhanced the uptake and utilization of palmitate in low-B12 hepatocytes for mitochondrial respiration, therefore, preventing the accumulation of lipids in hepatocytes.

### 3.5. Combined Treatment of Adiponectin and Metformin Ameliorates the Regulation of FA Synthesis and FAO in Low B12 Hepatocytes via Increased AMPK Activation

Since adiponectin treatment independently reduced lipid accumulation in low-B12 cells, we further assessed the effect of combined adiponectin–metformin treatment on lipid metabolism of hepatocytes with low B12. We previously observed that metformin alone had no effect on expression of AdipoR1 and R2 receptors in hepatocytes. However, the gene expression of adipoR1 and R2 receptors was significantly upregulated with low B12 (56.6% and 56.9%) compared with controls (28.3% and 26.8%, respectively) following the combined adiponectin–metformin treatment in hepatocytes (Figure 5A(i,ii)). Further to the upregulation in adipoR1 and R2, we subsequently assessed how this affects the phosphorylation of AMPK and ACC. We observed that the combined adiponectin–metformin treatment resulted in significant upregulation in pAMPKα and pACC levels (257.0% and 229.0%) with low B12 compared with controls (75.0% and 69.4%) (Figure 5B(i,ii)). This evidence suggests that the combined adiponectin–metformin treatment resulted in a significant activation of AMPKα and subsequent inactivation of ACC in low-B12 hepatocytes. Following the increased levels of activated AMPK (pAMPKα) in hepatocytes after the combined treatment with adiponectin and metformin, we further assessed the subsequent effect on lipogenesis. The expression genes regulating de novo FA synthesis was significantly downregulated with low B12, accounting for decreased expression of—SREBF1 (47.7%), ACC (40.0%), FASN (46.7%) and ELOVL6 (42.5%) (Figure 5C(i–iv), compared with controls [SREBF1 (41.8%), ACC (32.4%), FASN (45.6%) and ELOVL6 (40.2%) decrease] in the combined adiponectin–metformin-treated hepatocytes. Similarly, assessment of TG and cholesterol synthesis showed that, combined adiponectin–metformin treatment significantly downregulated the expression of genes regulating the synthesis of TG [SCD1 (49.7%), DGAT1 (45.6%) and DGAT2 (40.6%) decrease] (Figure 5D(i–iii)) and the master regulator of cholesterol LDLR (44.2%) (Figure 5E biosynthesis with low B12 than control [SCD1 (36.6%), DGAT1 (38.2%), DGAT2 (30.5%) and LDLR (31.1%) decrease]. To verify the impact of decreased lipogenesis resulting from the combined adiponectin–metformin treatment in low-B12 cells, the total intracellular triglyceride levels were subsequently quantified and assessed. We observed that the total intracellular TG level was significantly decreased (36.1%) with low B12 compared to controls (24.9 %) in the combined adiponectin-metformin-treated hepatocytes (Figure 5F). This evidence therefore suggests that the combined therapy reduced total TG levels significantly in low-B12 hepatocytes. To further validate the evidence of decreased expression of TG biosynthesis genes in low-B12 hepatocytes resulting from the combined adiponectin–metformin treatment, we further assessed the level of radiolabelled TGs synthesized following exposure of hepatocytes to radiolabelled long-chain fatty acid (^14^C-oleate) and L-carnitine. We observed that the level of radiolabelled TGs synthesized in hepatocytes was significantly decreased (49.1%) with low B12 compared with controls (38.5%) in the combined adiponectin–metformin-treated hepatocytes (Figure 5G). This therefore suggests that combined adiponectin–metformin treatment decreased the utilization of free long-chain FA (oleate) for TG synthesis in low-B12 hepatocytes.

Following the observation of significantly decreased lipogenesis via adiponectin–metformin co-treatment in low-B12 hepatocytes, we further assessed the effect on FAO. We observed that, the rate limiting enzyme CPT1a as well as ACADM were significantly upregulated, accounting for 68.8% (CPT1a) and 112.8% (ACADM) increase with low B12 compared with controls [45.1% (CPT1a) and 53.7% (ACADM)] in the combined adiponectin–metformin-treated hepatocytes (Figure 6A(i,ii)). This suggests that FAO in hepatocytes was significantly upregulated via adiponectin–metformin combined treatment in low-B12 cells.

To obtain further evidence in support of FAO, mitochondrial functional integrity was assessed using the extracellular seahorse XF24 flux assay. Following the combined adiponectin–metformin treatment, we observed that mitochondrial functional efficiency (OCR and SRC) of HepG2 in the presence of respiratory inhibitors (Oligomycin, FCCP and rotenone/antimycin) was significantly increased with low B12 (125%) compared with controls (77%) (Figure 6B(i,ii)). The basal uptake of long-chain FA (palmitate) by the cells after adiponectin–metformin co-treatment resulted in significantly increased functional efficiency (OCR and SRC) with low B12 (186%) compared with controls (109%) (Figure 6C(i–iii)). Lastly, functional efficiency of adiponectin–metformin-treated hepatocytes exposed to palmitate in the presence of the respiratory inhibitors was significantly higher with low B12 (75%) compared to controls (66%) (Figure 6D(i,ii)). This evidence shows that the efficiency of mitochondrial respiration utilizing palmitate in the substrate was significantly improved following combined adiponectin–metformin treatment, therefore, suggesting that accumulation of long-chain FA (palmitate) in low-B12 hepatocytes might be prevented by combined adiponectin–metformin treatment.

## 4. Discussion

The current study showed that in B12 deficient hepatocyte cell line HepG2, with or without metformin treatment, there was increased lipid accumulation and reduced fatty acid oxidation resulting from impaired activation and/or inactivation via phosphorylation of AMPK and ACC, respectively. We demonstrated that adiponectin independently expressed a significant lipid lowering effect in low-B12 cells, and its combined treatment with metformin, resulted in further lipid lowering effect in low-B12 cells via AdipoR1/R2-AMPK activation pathway.

Activation (phosphorylation) of AMPKα reduces lipid levels in the liver [26]. We observed decreased levels of pAMPKα in low B12 hepatocyte cell line, indicating that B12 independently plays a role in the activation of AMPK in the liver. B12 is a well-known source of one-carbon metabolites and methyl donors. Studies have shown that methyl donor supplementation in mice resulted in the activation of AMPK leading to reduction of hepatic lipid accumulation [27]. The precise mechanism underlying B12 activation of AMPK requires further investigation. Similarly, we observed that phosphorylated ACC was decreased in low-B12 cells. The inhibitory effect of AMPK on lipogenesis was reported to be achieved via phosphorylation (inactivation) of ACC [28]. This may confirm our observation that decreased pAMPKα with low B12 accounted for reduced pACC in hepatocytes. In addition, we observed that the regulation of pAMPKα and pACC in low-B12 cells, following metformin treatment, was impaired. Metformin treatment, according to previous studies involving rats and humans, resulted in a decrease of circulating B12 levels and was further suggested to account for altered B12 metabolism and distribution in tissues [29]. However, metformin has been shown to elevate phosphorylation of AMPK in the liver [30] via elevation of AMP:ATP and ADP:ATP ratios [31]. Another study in mice fed with a methyl donor deficient diet had decreased levels of AMP, ADP and ATP [32]. Our findings in support of these evidences, therefore, suggest that metformin activation of AMPK may be affected by methyl donor (B12) deficiency; however their role in AMP, ADP and ATP metabolism warrants further study.

We recently showed that B12 deficiency was associated with increased lipogenesis in hepatocytes [9] and adipocytes resulting in upregulation of genes regulating fatty acid and cholesterol biosynthesis [8], as well as higher TG levels. In this current study, the nuclear transcription factor SREBF1 and downstream genes regulating FAs, TG and cholesterol biosynthesis were upregulated with low B12. Studies reported that AMPK activation following metformin treatment, resulted in the phosphorylation and inactivation of SREBF1 leading to reduction in lipogenesis and hepatic accumulation of lipids. In this study, a similar effect was achieved in control cells with metformin but not in low-B12 cells. Several studies involving animal models and cell lines have reported the lipid lowering effect of metformin via reduction of SREBP1, ACC, FAS, ELOVL6, SCD1, HMGCS1 [33] and HMGCR [33] leading to reduction of intrahepatic triacylglycerol content [34]. However, in this study, regulation of these lipogenic genes by metformin was impaired in low-B12 cells. In addition, the total intracellular TG and radiolabelled TG synthesized in metformin treated low-B12 hepatocytes were significantly higher compared to controls. It therefore suggests that B12 deficiency might impair the desired lipid lowering effect of metformin in human hepatocytes. Evidence from clinical studies targeting the hepatic lipid lowering effect of metformin have also shown that intrahepatic TG level in human remains unaffected following metformin treatment [34]. The risk of B12 deficiency following metformin administration is associated with both dosage and duration (years) of treatment [35]. Hepatic deficiency of B12 may also occur after prolonged deprivation since the liver is able to maintain physiological B12 reserves up to five years [36]. In a rat model study, three weeks metformin treatment resulted in decline of circulating B12, whereas liver B12 content was significantly higher [29]. Therefore, prolonged metformin treatment is more feasible in humans than animal models leading to probable reduction in hepatic stores, and it explains the ineffective lipid lowering impact on intrahepatic TG levels in some humans could be due to hepatic B12 deficiency.

In a previous evidence, metformin reversed hepatic steatosis by blocking ACC via AMPK activation which subsequently minimized build-up of malonyl-CoA (inhibitor of CPT1), therefore, upregulating hepatic oxidation of fatty acids (FAO) [37]. In the current study, we observed that upregulation of the rate-limiting enzyme of FAO pathway, CPT1α, and downstream genes such as CACT, HADHA, ACADM, ACADS and ACADL were impaired with low B12 compared to controls. This may suggest that metformin did not significantly increase FAO in low-B12 cells compared to controls. Some studies involving humans [14], animals [38] and cell lines [39] have also shown that metformin failed to increase FAO in subjects. The inhibition of FAO in metformin-treated low-B12 cells may have resulted from low B12 inhibition of AMPK and ACC phosphorylation. Likewise, mitochondrial functional potency assessed by its spare respiratory capacity was compromised with low B12, which was not improved by metformin treatment. B12 deficiency is associated with T2DM and a study in diabetes cohort similarly showed that compromised hepatic and whole-body oxidation of FA was not improved after metformin treatment [40]. Our study therefore suggests that both FAO enzymes and mitochondrial functional efficiency in hepatocytes treated with metformin are hampered in low B12 (Figure 7A).

However, studies have demonstrated an association of adiponectin with decreased TG and fatty acid levels [41]. In the current study, adiponectin treatment increased the expression of adipoR1 and R2 receptors leading to higher activation of AMPK and pACC levels with low B12. However, the expression of adipoR1 and R2 have been shown to be significantly reduced due to methyl donors such as folate and choline deficiencies, respectively, although the underlying mechanisms requires further studies [42]. To the best of our knowledge, this is the first study to assess the effect of B12 on adiponectin receptors. In a mouse model, it has been shown that activation of AMPK with subsequent facilitation of FAO was achieved via binding of adiponectin to AdipoR1 and R2 [43], therefore, confirming our observation in this study.

Next, we observed that adiponectin treatment increased pAMPKα levels and resulted in elevation of pACC levels in low-B12 hepatocytes. AMPK is reported to induce inhibition of lipid biosynthesis through phosphorylation of ACC [44]. We further found that the master regulator of lipogenesis SREBF1 and downstream enzymes involved in de novo FA, TG and cholesterol synthesis were significantly decreased with low B12. Supporting this, a mice study showed suppression of SREBP1c by adiponectin via AdipoR1 and AMPK dependent pathways [45]. Similar studies have also shown that AMPK activation accounts for downregulation of genes involved in FA (ACC [46], FASN [46] and ELOVL6 [46]) triglyceride (SCD [46]) and cholesterol (LDLR [47]) synthesis. In this study, we also observed that adiponectin promoted FAO via increasing the expression of CPT1a and ACADM in low-B12 hepatocytes. Adiponectin, likewise, improved mitochondrial functional integrity in the utilization of substrates such as palmitate for energy metabolism. This is consistent with findings in skeletal muscles that adiponectin upregulated FAO via increasing expression of CPT1 [48] as well as elevated mitochondrial number and function, leading to increased oxidation of palmitate [49].

In line with the observation of adiponectin having the independent lipid lowering effect of adiponectin in low-B12 hepatocytes, we further assessed the combined effect of adiponectin and metformin on hepatic lipid metabolism. A clinical study in T2D and in vitro model of adipocytes treated with metformin have shown decreased levels of adiponectin [50]. Our experiments with combined adiponectin and metformin resulted in a significant increase in adipoR1/R2, pAMPKα and pACC with low B12. We also observed subsequent reduction in the expression of SREBF1 and the key enzymes involved in FA, TG and cholesterol synthesis under low B12 compared to basal control. An earlier study has showed that combination of adiponectin and metformin demonstrate a higher effect on blood glucose in animals [51]. Also, the combination of metformin with another potent activator of AMPK, lactoferrin, resulted in significant alleviation in the accumulation of lipids in the liver by increasing pAMPK/AMPK ratio in another study [52]. The authors also observed a significant reduction in FA (SREBP1, ACC, FAS) and cholesterol (HMGCR), therefore resulting in obesity prevention and improvement in metabolism of lipids [52]. Adiponectin is not expressed in the liver, but the circulatory adiponectin interacts with AdipoRs expressed in the liver leading to regulation of hepatic metabolism of lipids via AMPK and SREBP-1 signalling [53]. Activation of AMPK by adiponectin involves direct phosphorylation of threonine 172 (Thr172) located on the alpha subunit [53]. AMPK could also be indirectly activated by preservation of the alpha subunit from Thr172 phosphatase through allosteric modulation induced via high AMP/ATP ratio by metformin [54]. However, activation via the latter mechanism has been shown to be less than 5-fold [54], whereas combination of both phosphorylation and allosteric mechanisms accounts for >1000-fold activation of AMPK [54]. This may therefore explain our observation that activation of AMPK and pACC levels in low-B12 hepatocytes via adiponectin–metformin co-treatment could subsequently account for reduced hepatic lipid synthesis, upregulated oxidation of FA and improved mitochondrial functional integrity (Figure 7B). Evidence of both B12 deficiency and decreased adiponectin synthesis are reported in obesity, GDM, NAFLD and T2D patients on prolonged treatment with metformin. Clinically, women with gestational diabetes and obesity are on metformin treatment and if the drug is inefficient, these patients unknowingly having a B12 deficiency are given insulin, suggesting the patients are not responsive. However, our study implicates that if patients are non-responsive to metformin treatment, perhaps the subgroup of patients with low B12 levels should be monitored regularly, and the desired lipid lowering effect may be achieved via combination with adiponectin therapy rather than supplementation with B12. However, validation of this evidence in in vivo using animal and human models are recommended.

## 5. Conclusions

Low B12 hepatocytes in presence of metformin compromised lipid-lowering effect, whereas the desired lipid lowering effect of metformin was restored by combined adiponectin–metformin treatment. Our study evidences a novel regulatory mechanism of AMPK via adiponectin in lipid metabolism in hepatocytes and demonstrates that adiponectin-AMPK axis through adipoR1/R2 is a potential target during B12 deficiency and long-term metformin treatment.

## Figures and Tables

**Figure 1 nutrients-15-05046-f001:**
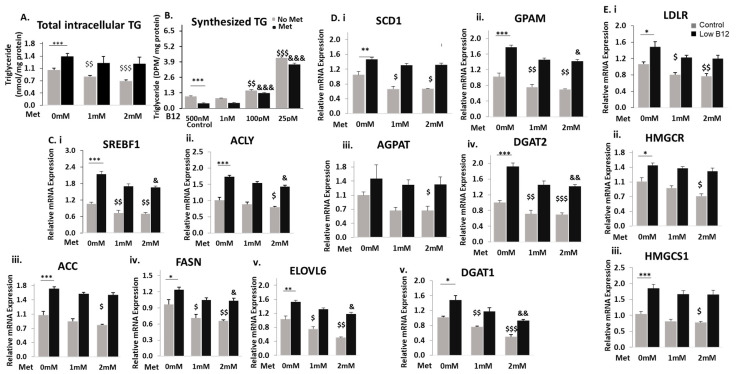
**Low B12 impedes the metformin action of lowering TG and expressing genes in fatty acid synthesis**. (**A**) Total intracellular levels of TG quantified in cells using the TG kit (ab65336) (Abcam Plc, Cambridge, UK) and normalized per milligram protein under each B12 condition. (**B**) Levels of synthesized TG in cells assessed with the radiochemical flux assay. HepG2 was first labelled with ^12^C-Oleate for 2 h, then followed by total lipid extraction, and the resultant radiolabelled triglyceride was separated on a thin-layer chromatography (TLC) plate with glyceryltripalmitate as standard and quantified with the scintillation counter Beckman coulter LS6500 (Beckman Coulter, Woonsocket, RI, USA) and normalized per milligram protein estimated with the Bradford method. (**C**) The mRNA expression of nuclear transcription factor SREBF1 (**i**) and enzymes regulating de novo fatty acid synthesis [ACLY (**ii**), ACC (**iii**) FASN (**iv**) and ELOVL6 (**v**)], (**D**) TG synthesis such as [SCD (**i**), GPAM (**ii**), AGPAT (**iii**), DGAT2 (**iv**) and DGAT1 (**v**)] and (**E**) cholesterol [LDLR (**i**), HMGCR (**ii**) and (HMGCS1 (**iii**)] normalized to 18S rRNA endogenous control (Applied Biosystems, Knutsford, UK). The data is representative of mean ± SEM (*n* = 6), and * represents significance compared to controls B12 (500 nM) and low B12 (25 pM), $—compared to controls B12 (500 nM) in 1 mM and 2 mM metformin and &—compared to low B12 (25 pM) with 1 mM metformin and 2 mM, respectively; * *p* < 0.05, ** *p* < 0.01, *** *p* < 0.001; $ < 0.05, $$ < 0.01, $$$ < 0.001; & < 0.05, && < 0.01, &&& < 0.001.

**Figure 2 nutrients-15-05046-f002:**
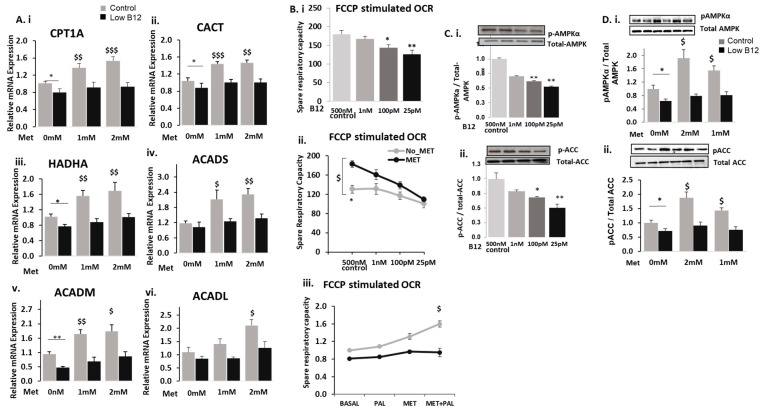
**Low B12 impedes metformin action on fatty acid oxidation (FAO) and mitochondrial function via impaired AMPK:** The mRNA expression of (**A**) fatty acid oxidation genes such as CPT1 (**i**), CACT (**ii**), HADHA (**iii**), ACADS (**iv**), ACADM (**v**) and ACADL (**vi**) normalized to 18S rRNA endogenous control (Applied Biosystems, Knutsford, UK). (**B**) The spare respiratory capacity of hepatocytes in: (**i**) substrate-rich KHB substrate medium containing glucose (2.5 mM), pyruvate (1 mM), an amino acid (L-Glutamine) (2 mM) and BSA (0.1%) at pH 7.4 under various B12 conditions; (**ii**) in metformin-treated hepatocytes under various B12 conditions and (**iii**) in a limited-substrate KHB medium containing only 0.5 mM L-carnitine and 1.25 mM glucose and supplemented with palmitate (200 µM)/BSA (33.3 µM) (basal control) under different B12 conditions. (**C**) Represents pAMKα (**i**) and pACC (**ii**) levels normalized to total AMPK and ACC, respectively. (**D**) Metformin treatment failed to restore pAMPKα (**i**) and pACC (**ii**) levels with low B12. The data is representative of mean ± SEM (*n* = 6), and * represents significance compared to controls B12 (500 nM) and low B12 (25 pM), $—compared to controls B12 (500 nM) in 1 mM and 2 mM metformin and &—compared to low B12 (25 pM) with 1 mM metformin and 2 mM respectively; * *p* < 0.05, ** *p* < 0.01, $ < 0.05, $$ < 0.01, $$$ < 0.001.

**Figure 3 nutrients-15-05046-f003:**
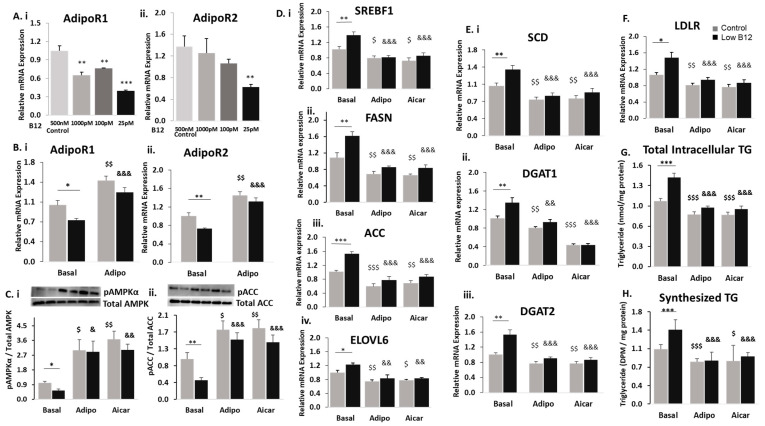
Adiponectin ameliorates the regulation of FA synthesis in low-B12 hepatocytes via increased AMPK activation and AdipoR1/R2 (**A**) The mRNA expression of AdipoR1 (**i**) and AdipoR2 (**ii**) under low B12 conditions and (**B**) mRNA expression of AdipoR1 (**i**) and AdipoR2 (**ii**) following 24 h treatment with 25 ng adiponectin. (**C**) Protein levels of pAMPKα (**i**) and pACC (**ii**) normalized to total AMPK and ACC, respectively in adiponectin [or AICAR (positive control)] treated hepatocytes. (**D**) mRNA expression of enzymes nuclear transcription factor SREBF1 (**i**) and enzymes regulating FA synthesis, such as FASN (**ii**), ACC (**iii**), ELOVL6 (**iv**); (**E**) triglyceride synthesis, such as SCD (**i**), DGAT1 (**ii**) and DGAT2 (**iii**); and (**F**) cholesterol synthesis, such as LDLR normalized to 18S rRNA endogenous control (Applied Biosystems, Knutsford, UK). (**G**) Total intracellular levels of triglycerides quantified in hepatocytes using the TG kit (ab65336) (Abcam Plc, Cambridge, UK) and normalized per milligram protein under each B12 condition. (**H**) Levels of synthesized TG in hepatocytes assessed with the radiochemical assay and quantified as disintegrations per minute (DPM) using the scintillation counter and normalized per milligram (mg) protein. The data is representative of mean ± SEM (*n* = 6), * represents significance compared to controls B12 (500 nM) and low B12 (25 pM), $—compared to controls B12 (500 nM) with adiponectin or aicar and &—compared to low B12 (25 pM) with adiponectin or aicar, respectively; * *p* < 0.05, ** *p* < 0.01, *** *p*< 0.001; $ < 0.05, $$ < 0.01, $$$ < 0.001; & < 0.05, && < 0.01, &&& < 0.001.

**Figure 4 nutrients-15-05046-f004:**
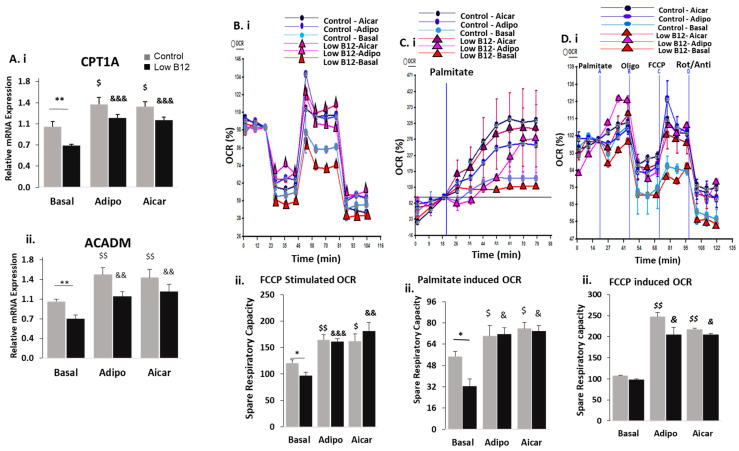
**Adiponectin ameliorates the regulation of FAO in low-B12 hepatocytes via increased AMPK activation**: (**A**) The mRNA expression of enzymes regulating FAO pathway, such as CPT1a (**i**) and ACADM (**ii**), were normalized to 18S rRNA endogenous control (Applied Biosystems, Knutsford, UK). (**B**) (**i**) Oxygen consumption rate (OCR) and (**ii**) spare respiratory capacity following treatment with 25ng adiponectin (or 1 mM AICAR, positive control) in hepatocytes incubated with a rich-substrate KHB medium containing glucose (2.5 mM), pyruvate (1 mM), an amino acid (L-Glutamine) (2 mM) and BSA (0.1%) at pH 7.4 under various B12 conditions. (**C**) (**i**) Oxygen consumption rate (OCR) and (**ii**) spare respiratory capacity following injection with palmitate (200 µM)/BSA (33.3 µM) (basal control) in 25 ng adiponectin (or 1 mM AICAR, positive control)-treated-hepatocytes in a limited-substrate KHB medium containing only 0.5 mM L-carnitine and 1.25 mM glucose under different B12 conditions. (**D**). (**i**). Oxygen consumption rate (OCR) and (**ii**) spare respiratory capacity in hepatocytes exposed to palmitate (200 µM) and respiratory inhibitors (Oligomycin, FCCP and rotenone/antimycin). The data is representative of mean ± SEM (*n* = 6), and * represents significance compared to controls B12 (500 nM) and low B12 (25 pM), $—compared to controls B12 (500 nM) with adiponectin or aicar and &—compared to low B12 (25 pM) with adiponectin or aicar, respectively; * *p* < 0.05, ** *p* < 0.01; $ < 0.05, $$ < 0.01; & < 0.05, && < 0.01, &&& < 0.001.

**Figure 5 nutrients-15-05046-f005:**
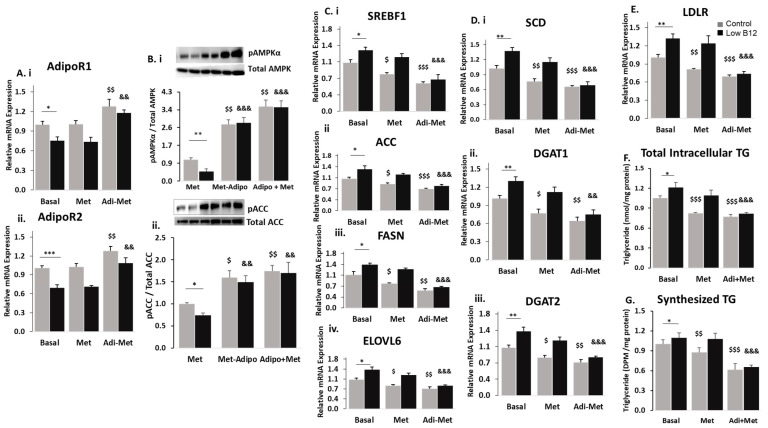
**Combined treatment of adiponectin and metformin ameliorates the regulation of FA synthesis in low-B12 hepatocytes via increased AMPK activation**: (**A**) The mRNA expression of AdipoR1 (**i**) and R2 (**ii**) and master regulator of lipogenesis SREBF1 (**C**) (**i**) as well as enzymes regulating fatty acid synthesis, including ACC (**ii**), FASN (**iii**) and ELOVL6 (**iv**), TG synthesis (**D**), including SCD (**i**), DGAT1 (**ii**) and DGAT2 (**iii**) and cholesterol LDLR (**E**), were normalized to 18S rRNA endogenous control (Applied Biosystems, Knutsford, UK). (**B**) Protein levels of pAMPKα (**i**) and pACC (**ii**) following combined or sequential treatment of adiponectin and metformin and normalized to total AMPK and ACC, respectively. (**F**) Total intracellular levels of triglycerides quantified in hepatocytes using the TG kit (ab65336) (Abcam Plc, Cambridge, UK) and normalized per milligram protein under each B12 condition. (**G**) Levels of synthesized TGs in hepatocytes assessed with the radioactive flux assay and quantified as disintegrations per minute (DPM) using the scintillation counter and normalized per milligram (mg) protein. The data is representative of mean ± SEM (*n* = 6), and * represents significance compared to controls B12 (500 nM) and low B12 (25 pM), $—compared to controls B12 (500 nM) in metformin only or both metformin and adiponectin and &—compared to low B12 (25 pM) in both metformin and adiponectin, respectively; * *p* < 0.05, ** *p* < 0.01, *** *p* < 0.001; $ < 0.05, $$ < 0.01, $$$ < 0.001; && < 0.01, &&& < 0.001.

**Figure 6 nutrients-15-05046-f006:**
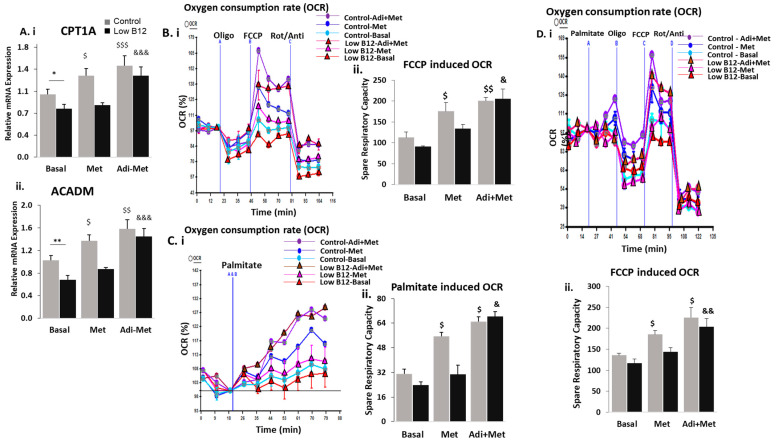
**Combined treatment of adiponectin and metformin ameliorates the regulation of FAO in low-B12 hepatocytes via increased AMPK activation**: The mRNA expression of key enzymes regulating FAO (**A**) such as CPT1 (**i**) and ACADM (**ii**) normalized to 18S rRNA endogenous control (Applied Biosystems, Knutsford, UK). (**B**) (**i**) Oxygen consumption rate (OCR) and (**ii**) spare respiratory capacity following combined treatment with 25 ng adiponectin and 2 mM metformin in hepatocytes incubated with a rich-substrate KHB medium containing glucose (2.5 mM), pyruvate (1 mM), amino acid (L-Glutamine) (2 mM) and BSA (0.1%) at pH 7.4 under various B12 conditions. (**C**) (**i**) Oxygen consumption rate (OCR) and (**ii**) spare respiratory capacity following injection with palmitate (200 µM)/BSA (33.3 µM) (basal control) in adiponectin and metformin-treated hepatocytes in a limited-substrate KHB medium containing only 0.5 mM L-carnitine and 1.25 mM glucose under different B12 conditions. (**D**) (**i**) Oxygen consumption rate (OCR) and (**ii**) spare respiratory capacity in hepatocytes exposed to palmitate (200 µM) and respiratory inhibitors (Oligomycin, FCCP and rotenone/antimycin). The data is representative of mean ± SEM (*n* = 6), and * represents significance compared to controls B12 (500 nM) and low B12 (25 pM), $—compared to controls B12 (500 nM) in only metformin or both metformin and adiponectin and &—compared to low B12 (25 pM) with both metformin and adiponectin, respectively; * *p* < 0.05, ** *p* < 0.01; $ < 0.05, $$ < 0.01, $$$ < 0.001; & < 0.05, && < 0.01, &&& < 0.001.

**Figure 7 nutrients-15-05046-f007:**
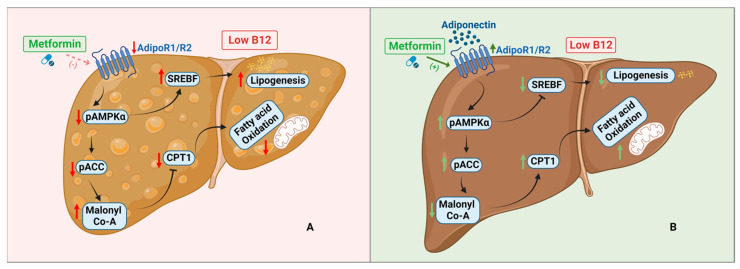
**Schematic representation of hepatocytes with low B12 status upon metformin and adiponectin treatment**: (**A**) Following metformin treatment (dotted line—less activation) with low B12, pAMPKa and pACC levels are decreased with subsequent increase in the levels of SREBF and malonyl-CoA that further inhibits CPT1. In effect, lipogenesis is upregulated whereas fatty acid oxidation (FAO) is decreased, resulting in accumulation of lipids in the hepatocytes. (**B**) Adiponectin and metformin (solid line—activation) together results in the activation of AMPK. Activated AMPK directly phosphorylates ACC which results in the inhibition of malonyl-CoA. Decreased malonyl-CoA levels facilitate the upregulation of CPT1 and subsequently improves mitochondrion function with increased oxidation of fatty acids (FAO). Similarly, activation of AMPK results in the inactivation of SREBF, which subsequently decreases lipogenesis in hepatocytes (created with BioRender.com).

## Data Availability

All data would be made available by corresponding author upon request.

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
