# Peer review of "Metformin Inefficiency to Lower Lipids in Vitamin B12 Deficient HepG2 Cells Is Alleviated via Adiponectin-AMPK Axis"

_nutrients, 2023, doi:10.3390/nu15245046_

Round 1

Reviewer 1 Report

Comments and Suggestions for Authors

This manuscript "Metformin inefficiency to lower lipids in vitamin B12 deficient hepatocytes is alleviated via Adiponectin-AMPK axis ", based on the background of the increasing global epidemic of obesity and the increasing incidence of corresponding complications, the background of vitamin B12 deficiency will affect the regulation of metformin on liver lipid metabolism. Activation of Adiponectin-AMPK axis was found to restore the regulatory effect of metformin on liver lipid metabolism. In conclusion, the article focuses on the current epidemic diseases and treatment phenomena and proposes improvement methods. Since this article does not have any graphic results, it is impossible to judge whether your experimental results prove your point of view, nor how reliable the experimental results are. It is recommended to add the graphic results of the experimental results to the article before considering modification or acceptance.

Questions were raised as below and needed to be addressed.

1. There is no chart showing any experimental results in the paper, which needs to be added.

2. Due to the complexity of the mechanism study in this paper, which involves Adiponectin-AMPK axis, it would be more helpful to understand this process if organic maps can be drawn in this paper to show how the activation of this axis can restore the effect of metformin.

Reviewer 2 Report

Comments and Suggestions for Authors

In this study, the authors aim to explore two key questions: firstly, whether low Vitamin B12 levels impair the effectiveness of metformin on hepatic lipid metabolism, and secondly, whether activating the AMPK-adiponectin axis mitigates metformin's effects in a low B12 state. However, I have several concerns and questions about the article that I wish to express:

Major Comments:

1. The introduction states that metformin does not affect intrahepatic triglyceride (TG) levels. This assertion seems inconsistent with the experimental findings presented.

2. What is the rationale behind the chosen Vitamin B12 dosages in the cell culture studies? Is there a significant difference in dosage between the control group and the low B12 group?

3. The article's clinical relevance should be considered. Long-term metformin use can lead to Vitamin B12 deficiency, and while B12 supplementation is relatively safe and reliable, the authors suggest that combining treatment with adiponectin might be more effective for lipid reduction. Could you provide further explanation on this?

4. This research potentially overcomplicates straightforward questions. Proposing adiponectin as a treatment introduces several issues, including appropriate dosage, indications, and contraindications, which may not have practical clinical significance.

5. It is recommended to include more in-vivo animal studies to strengthen the evidence base.

Minor Comments:

1. Including charts and graphs could enhance readers' comprehension of the study.

2. In line 57, the text reads, “However, few studies in animals and a majority (50%) in humans have shown that metformin treatment had no effect on intra-hepatic TG content.” This statement is somewhat unclear.

3. Please use full names when abbreviations are first introduced in the text.

Reviewer 3 Report

Comments and Suggestions for Authors

The authors show that in hepatocytes exposed to low vitamin B12 with metformin treatment, there is a trend for reduction in the expression of a pool of genes. Is this a specific effect for genes responsible for lipogenesis and FA synthesis or is this a non-specific effect due to very well-known roles of vitamin B12 in DNA/RNA synthesis. Were other genes not related to lipid metabolism investigated, as negative controls?

In their work, the authors extensively used mRNA expression for multiple genes as a read-out. Their major conclusions came mainly from RNA expression data. While for multiple experiments the statistical significance was reached, what it the real biological significance? For example, in figure 1.D.i the relative mRNA expression for SCD1 the low B12 vs control in no Metformin treatment increases from 1 to 1.5. Such a modest change in mRNA expression does not necessary translate to changes in protein translation, or changes in protein function. The authors must include, at least for the most relevant genes, proteomic analysis (e.g., immunoblot.)

While HepG2 has been used by many groups to study hepatocytes function, at the end of the day they are still cancerous-origin cells with significant genome instability and multiple aberrant metabolic pathways. Therefore, to claim their conclusions, the authors should either repeat critical experiments using normal hepatocytes, or to clearly underline that their data were obtained using one cancer cell line. This will include obviously adding the “HepG2” in the title of the manuscript.

Hep2G source, authenticity and mycoplasma assays must be included.

Round 2

Reviewer 1 Report

Comments and Suggestions for Authors

No more comments 

Reviewer 3 Report

Comments and Suggestions for Authors

I would like to thank the authors for their pertinent answers and additional data. I believe that in the present form the manuscript is suitable for publication.

Thank you.